# Effective *Chlorella vulgaris* Biomass Harvesting through Sulfate and Chloride Flocculants

**Wei Ma** [1,2], **Chenchen Feng** [3], **Fachun Guan** [2], **Dianrong Ma** [1,*] **and Jinling Cai** [3,*]

1 Rice Research Institute, Shenyang Agricultural University, Key Laboratory of Northern Japonica Super Rice Breeding, Ministry of Education, Shenyang 110866, China
2 Jilin Academy of Agricultural Sciences, Changchun 130033, China
3 College of Chemical Engineering and Materials Science, Tianjin University of Science & Technology, Tianjin Key Laboratory of Brine Chemical Engineering and Resource Eco-utilization, Tianjin 300457, China
* Correspondence: madianrong@syau.edu.cn (D.M.); jinlingcai@tust.edu.cn (J.C.); Fax: +86-22-60601305 (J.C.)

**Abstract:** Efficient microalgae harvesting is a great challenge hindering diverse industrial applications of microalgae. Flocculation is regarded as an effective and promising technology for microalgae harvesting. In this study, sulfate ($Al_2(SO_4)_3$ and $Fe_2(SO_4)_3$) and chloride flocculants ($AlCl_3$ and $FeCl_3$) were used to harvest *Chlorella vulgaris*. Flocculation conditions, including flocculant dose, flocculation time, stirring speed, stirring time, and flocculation pH, were optimized, and flocculant effects on microalgal cell status, floc characteristics, biomass composition, algal cell re-culture, and media recycling were investigated. All flocculants exhibited efficient flocculation efficiency (93.5–98.8%) with lower doses of sulfate salts (60 mg/L algal culture) and higher doses of chloride salts (100 mg/L algal culture). The tested flocculants had no obvious influence on biomass composition (including lipids, carbohydrates, proteins, and carotenoids), and microalgal cells in flocs could efficiently regrow. The spent medium of all treatments was successfully recycled for subsequent cell growth, thus reducing dependency on fresh medium.

**Keywords:** microalgae; harvest; flocculation; biomass recovery; culture medium recycling

## 1. Introduction

Reactive microalgae which are regarded as promising and sustainable cell factories are rich in high-value biocompounds, including zeaxanthin, β-carotene, lutein, agar, enzymes, polyunsaturated fatty acids, vitamins, amino acids, and various nutraceutical and pharmaceutical substances [1]. Before these high-value biocompounds are extracted from microalgae, an appropriate harvesting method is needed to concentrate the biomass. However, microalgae harvesting is still a constraint, due mainly to the density of marginal growth media (approximately 1.02 g/L), tiny veritable cell size (2–20 μm), and negative charge (from −7.5 to −40 mV), which prevent them from clumping together [2]. Harvesting costs account for approximately 90% of the cost of biomass production instruments [3], and harvesting cost is approximately 30% of the total microalgal production cost [4].

Microalgae harvesting is an economical key for commercial microalgae production, and selecting suitable harvesting technology can increase biomass production and reduce the overall production cost. At present, the main recovery technologies are centrifugation, filtration, flotation, and flocculation. Among these methods, centrifugation is currently used in industry because of its high efficiency close to 100%. However, centrifugation has the limitations of high energy consumption as well as operating cost [5] and mainly used for harvesting microalgae for high-value bioproducts. The main disadvantages of filtration are pollution and clogging, and the colloidal stability of algal cells limits the efficiency of filtration [2]. In addition, the sedimentation process takes a long time. Flocculation is a superior method because it is effective, efficient, low-cost in large-scale use, and suitable for a variety of microalgae species [2].

Flocculation is achieved by adding chemicals or changing the environment to overcome the natural repulsion of microalgae and make them combine into aggregates and settle. In general, the function of flocculants is to counteract the negative charge of microalgae and promote the self-aggregation of microalgae [2]. Flocculants can be divided into inorganic flocculants, including metal ions such as $Al^{3+}$, $Fe^{3+}$, $Ca^{2+}$, and $Mg^{2+}$ [6,7], and organic polymer flocculants such as chitosan and polyacrylamide [7,8]. Flocculant type determines the flocculation efficiency. Inorganic flocculants are widely used in microalgae harvesting with low price but high dosage requirements [6,8]. In contrast, organic flocculants at low dosages tend to have higher harvesting efficiency, but the price is expensive [7,8]. Different flocculants showed different flocculation efficiency on freshwater and marine microalgal strains, which is listed in Table 1.

If there is no strict demand for the supernatant, inorganic flocculants are a good choice. Many studies have been carried out on the use of inorganic flocculants to collect microalgae [7]; in particular, chloride and sulfate salts have high solubility and a wide concentration range and are very effective for harvesting microalgae. Some researchers have found that chlorides are more effective than sulfate salts in harvesting microalgae [9,10], while others point out that sulfates are more effective than chlorides [11,12]. In addition, some researchers have not observed statistically significant differences between chlorides and sulfates [6,13]. However, there have been no studies comparing chlorides and sulfates with the same metal ions, such as $Fe^{3+}$ and $Al^{3+}$. Therefore, this study attempted to compare the efficiency of chloride and sulfate in harvesting microalgae.

The main objective of microalgae harvesting Is to obtain concentrated algal biomass for downstream processing. The flocculants combined with microalgae may pollute the algal biomass and affect the subsequent treatment. However, the effects of flocculants on downstream processes are not clear, especially the state of cells in flocs, the regeneration ability of cells in flocs, and the extraction of compounds from biomass. In addition, water reuse is another potential way to reduce the costs of microalgae production. Recovering the waste medium after microalgae harvesting can reduce water demand by 80%, nutrient inputs by 44%, and subsequent wastewater treatment [14,15]. A few studies have been conducted on the regeneration of microalgae from flocculated and sterilized or filtered waste media such as *Scenedesmus* sp., *C. vulgaris*, *C. fusca*, *Muriellopsis* sp., *Scenedesmus subspicatus*, and *Nannochlropsis oculate* [16–18]. However, there are few studies on the recoverability of waste media (unpurified) from microalgae harvested by flocculants for subsequent cultivation of microalgal cells. In this study, the flocculation performance of *C. vulgaris* was evaluated using two types of chloride flocculants ($AlCl_3$ and $FeCl_3$) and two types of sulfate salts ($Al_2(SO_4)_3$ and $Fe_2(SO_4)_3$). Furthermore, the reusability of spent medium for culturing microalgae was assessed, and the impacts of flocculants on the quality of the harvested microalgae were also examined.

The objectives of this work are as follows: (I) The optimal flocculation conditions of *Chlorella vulgaris* with $Al_2(SO_4)_3$, $AlCl_3$, $Fe_2(SO_4)_3$ and $FeCl_3$ as flocculants will be obtained in a single-factor experiment. (II) Microalgal cell status and floc characteristics between raw cells and flocculated cells will be compared in this article. (III) Variation in *Chlorella vulgaris* content after flocculation will be shown in this study, including total lipids, carbohydrates, proteins, and carotenoids. (IV) The regrowth of settled flocs and reuse of spent medium will be tested to elucidate the effects of flocculants on downstream processes.

**Table 1.** Flocculation efficiency of different flocculants on different freshwater and marine strains.

| Flocculant Name | Microalgae Species | Marine/ Freshwater | Flocculation Conditions | Flocculation Efficiency | References |
|---|---|---|---|---|---|
| $Al_2(SO_4)_3$ | *N. salina* (CCAP849/3) | marine | flocculant dosage: 114.4 mg/L | 95% | [19] |
| $Al_2(SO_4)_3$ | *N. salina* (CCAP849/3) | marine | flocculant dosage: 114.4 mg/L; feed flow rate: 56.5 mL/min | 40–50% | [19] |
| $Al_2(SO_4)_3$ | *N. salina* (CCAP849/3) | marine | flocculant dosage: 229 mg/L; feed flow rate: 56.5 mL/min | 75% | [19] |
| $Al_2(SO_4)_3$ | *N. salina* (CCAP849/3) | marine | flocculant dosage: 229 mg/L; feed flow rate: 20.4 mL/min | $(86.1 \pm 0.1)\%$ | [19] |
| $FeCl_3$; $Al_2(SO_4)_3$ | *C. minor* | freshwater | initial biomass concentration: 1.0 g/L; $FeCl_3$ 250 mg/L or $Al_2(SO_4)_3$ 275 mg/L | >95% | [20] |
| cationic polyacrylamide polymer | *Chlorella vulgaris* (CS-41) | freshwater | | >97% | [21] |
| poly (c-glutamic acid) | *C. vulgaris*; *C. protothecoides* | marine; freshwater | c-PGA dosage: 22.03 mg/L; biomass concentration: 0.57 g/L; salinity: 11.56 g/L | 91%; 98% | [22] |

## 2. Materials and Methods

### 2.1. Microalgae Strain and Culture Conditions

*C. vulgaris* (FACHB-275) was purchased from Freshwater Algae Culture Collection at the Institute of Hydrobiology (Wuhan, China). The microalgae were preserved and grown in BG11 medium according to Pandey et al. [17]. The initial pH, light intensity, and temperature were maintained at $7.00 \pm 0.2$, $65 \pm 5$ μmol/m$^2$.s, and $25 \pm 1$ °C, respectively. Floc forming pic is shown in Figure 1.

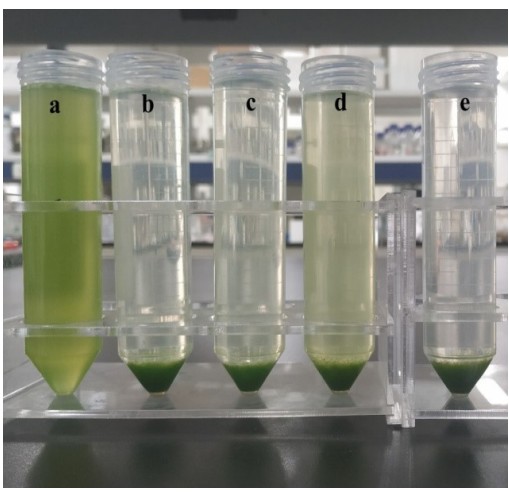

**Figure 1.** Flocculation formed by different flocculants (a: raw; b: $Al_2(SO_4)_3$; c: $AlCl_3$; d: $Fe_2(SO_4)_3$; e: $FeCl_3$).

### 2.2. Flocculation Experiment

Before flocculation, the C. vulgaris suspension was diluted with BG11 medium to an $OD_{680}$ (optical density measured at 680 nm) of 1.2 (approximately 708 mg dry weight/L) to obtain comparable initial conditions between different runs. For comparison, the effects of flocculant types ($Al_2(SO_4)_3$, $AlCl_3$, $Fe_2(SO_4)_3$, and $FeCl_3$), flocculant dose (sulfate dose 20–100 mg/L around the optimal dose 60 mg/L; chloride dose 60–140 mg/L around the optimal dose 100 mg/L), flocculation time (10–70 min), stir speed (100–500 rpm), stir time (1–5 min), and pH values (4–10) on flocculation efficiencies of C. vulgaris were studied through a one-factor method by varying one parameter and retaining the other factors as constant. C. vulgaris cells were mixed using a magnetic stirrer to ensure complete dispersal of the flocculants after adding the flocculant.

### 2.3. Evaluation of Flocculation Efficiency

To calculate flocculation efficiency, microalgae samples were collected before and after flocculation from the middle of the clarified zone, and the $OD_{680}$ was measured using a spectrophotometer (UV-1800, Shimadzu, Kyoto, Japan). The flocculation efficiency was determined using the following equation [23]:

$$\text{Flocculation efficiency (\%)} = 100\% \times (OD_0 - OD_t)/OD_0 \tag{1}$$

where $OD_0$ and $OD_t$ are the absorbances of the initial microalgal culture and supernatant after flocculation at 680 nm, respectively.

### 2.4. Reculture of Microalgal Cells in Flocs and Recycling of Spent Medium

After flocculation, the settled flocs and cultivation medium were separated. The flocs were filtered using filter paper (Beimu, Hangzhou, China). The flocs with microalgal cells that remained on the filter paper were washed using BG11 medium, stirred into single algal cells, and re-cultured in fresh BG11 medium. Fresh algal culture was used as a control, and the growth of the recovered *C. vulgaris* was compared with the control, in which the initial $OD_{680}$ was adjusted to 0.2 to obtain greater growth space.

The residual culture media after removing flocs was used as "spent medium". The spent medium pH and nutrient concentration were adjusted to standard BG11 medium, and fresh algal cells were inoculated into the spent medium. The control comprised fresh BG11 medium, and the growth of the algal cells in the spent medium and fresh medium was compared. The growth conditions for *C. vulgaris* were the same as the growth conditions described earlier (see Section 2.1).

### 2.5. Microscopic Images

The *C. vulgaris* cells before and after harvesting were observed under a brightfield microscope at 400× magnification (H550S Nikon, Tokyo, Japan). To capture the microstructure differences between the cell surfaces before and after harvesting, scanning electron microscopy (SEM, JSM-6380LV, Agilent, Tokyo, Japan) was employed.

### 2.6. Analysis Methods and Statistical Analysis

Microalgal biomass density was monitored by measuring the absorbance at 680 nm. The cell dry weight was measured as follows. The microalgal suspension was filtered, washed with distilled water to remove all the absorbed salts, and dried at 60 °C to constant weight. A calibration curve between OD values and corresponding cell dry weight was obtained (cell dry weight (g/L) = $0.59 \times OD_{680\,nm}$, $R^2 = 0.9949$). The carbohydrate content was determined according to Haldar et al. [24], protein composition was determined by the Coomassie Brilliant Blue method [25], lipid content was determined according to Bligh and Dyer [26], and carotenoid content was measured according to Wellburn [27].

All flocculation tests and measurements were repeated in triplicate. The collected data were subjected to Tukey's test through one-way analysis of variance (ANOVA) ($p < 0.05$),

with flocculant types, flocculant dose, stirring speed, stirring time and pH values as the sources of variation. Statistical analysis was performed using Prism version 24 (IBM SPSS Statistics). Data are shown as the mean values $\pm$ SD of three replicates.

## 3. Results and Discussion

### 3.1. Flocculants Dose

A dose–response relationship was observed when the flocculant was used individually (Figure 2a). In addition, the flocculation efficiency of *C. vulgaris* increased to its optimum point with increasing flocculant dosage. $Al^{3+}$ or $Fe^{3+}$ with a positive charge, can neutralize the negative charge on the surface of *C. vulgaris* and overcome static stability [4], and the increase in ionic charge efficiency is in direct proportion to the increase in flocculant dose.

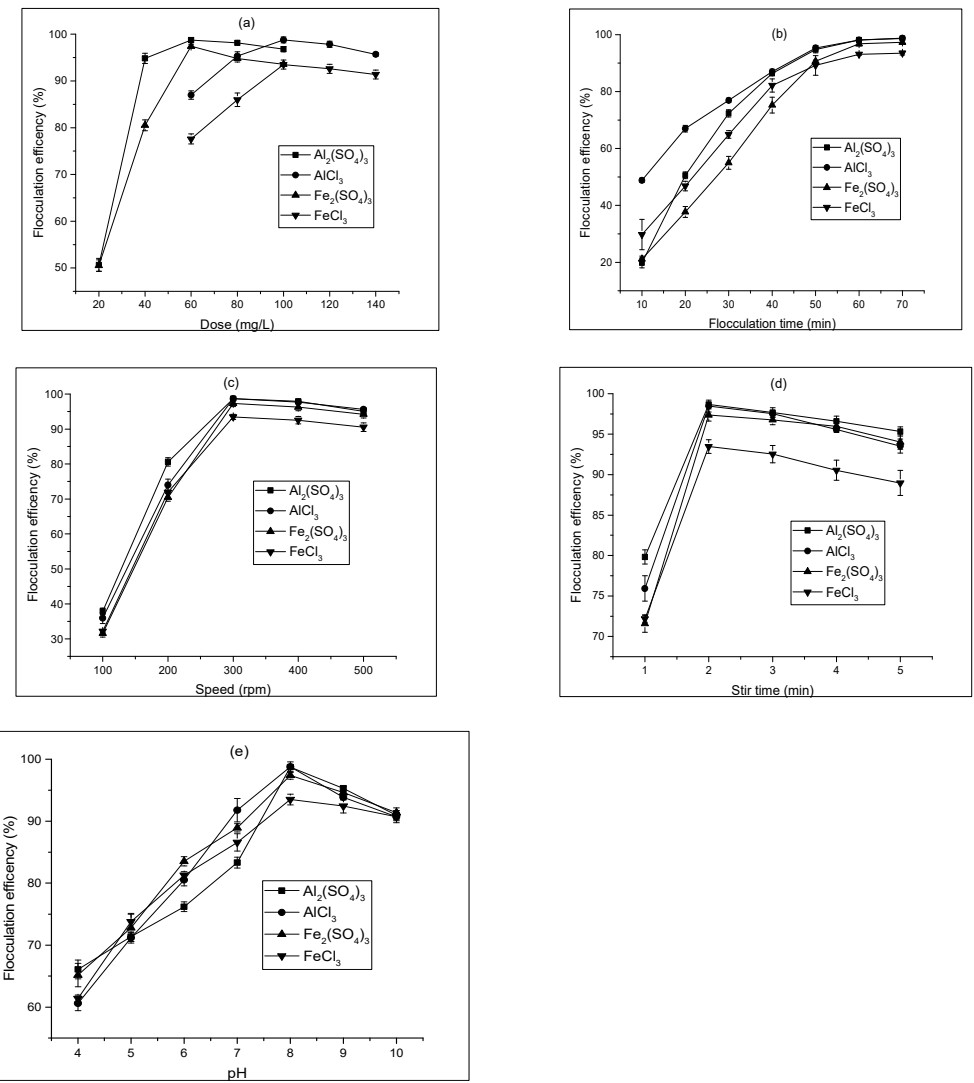

**Figure 2.** (**a**) Flocculation efficiency of different doses of flocculants. Condition: stirring for 2 min at 300 r/min, pH 8.0, settled for 70 min. (**b**) Flocculation efficiency at different settlement times. Condition: stirring for 2 min at 300 r/min, pH 8.0, 60 mg/L sulfate and 100 mg/L chloride. (**c**) Flocculation efficiency at different stirring speeds. Condition: stirring for 2 min at pH 8.0, settled for 60 min, 60 mg/L sulfate and 100 mg/L chloride. (**d**) Flocculation efficiency at different stirring times. Condition: stirring 300 r/min, pH 8.0, settled for 60 min, 60 mg/L sulfate and 100 mg/L chloride. (**e**) Flocculation efficiency at different pH values. Condition: stirring for 2 min at 300 r/min, settled for 60 min, 60 mg/L sulfate and 100 mg/L chloride. Data are reported with $+/-$ SD, and n = 3.

Maximum harvesting efficiencies of 98.73 ± 0.49% and 97.43 ± 0.42% were achieved using 60 mg/L sulfates ($Al_2(SO_4)_3$ and $Fe_2(SO_4)_3$), respectively, whereas flocculation efficiencies of 98.77 ± 0.64% and 93.50 ± 0.96% were attained using 100 mg/L $AlCl_3$ and $FeCl_3$, respectively. The optimum dose of sulfates (60 mg/L $Al_2(SO_4)_3$ and $Fe_2(SO_4)_3$) was relatively lower than the optimum dose of chlorides (100 mg/L $AlCl_3$ and $FeCl_3$). Similar findings have been reported by Reyes and Labra [11], who demonstrated that a low dose of sulfate (1.5 g/L $Al_2(SO_4)_3$) and a high dose of chloride (7.0 g/L $FeCl_3$) resulted in *Scenedesmus* sp. flocculation efficiency of 99.4% and 98.4%, respectively. However, some researchers have reported that the flocculation efficiency of low-dose chlorides is comparable to or higher than the flocculation efficiency of high-dose sulfates [9,10], while some studies have shown that there is no statistically significant difference in flocculation efficiency between chloride and sulfate, such as *Phaeodactylum tricornutum* [6], and a marine microalgae species, *Isochrysis galbana* [13]. This difference in the optimum dosage of flocculants in these studies may be due to changes in microalgae strains, cell wall composition and growth conditions.

In this study, further increasing the dosage of flocculant did not improve the flocculation efficiency. When flocculant concentration reached certain value, the metal cations completely neutralized the surface negative charge on the microalgal cells [7]. A further increase in the dose of flocculant will lead to a positive charge on the cell surface, increase the repulsion force between cells, stabilize the microalgae suspension and reduce the flocculation efficiency [28].

From Figure 2a, we can conclude that the flocculation activity of aluminium salts is higher than the flocculation activity of ferric salts, whether sulfate or chloride. This phenomenon can be explained by the following two points. First, aluminium salts have a higher cation concentration than ferric salts in both sulfate and chloride for an equal mass concentration. Second, this conclusion may be explained by the fact that the surface charge densities of $Al^{3+}$ and $Fe^{3+}$ are 95.5 $nm^{-2}$ and 58.0 $nm^{-2}$ based on the 0.050 nm radius of $Al^{3+}$ and 0.064 radius of $Fe^{3+}$, respectively. $Al^{3+}$ has a higher surface charge density than $Fe^{3+}$, which likely improves its ability to bridge cells and neutralize the surface charge [20].

The residual ion concentration is known to increase with increasing flocculant dose. In this study, the optimal concentrations of sulfates and chlorides were 60 and 100 mg/L, respectively, which are lower than the optimal concentration in most reports [11,13,28]. The low dosage of flocculants can reduce the content of residual ions that may contaminate the medium and damage cell vitality. Furthermore, a low flocculant dosage can reduce the cost of the flocculation process.

### 3.2. Flocculation Time

The impact of flocculation time on *C. vulgaris* flocculation efficiency is shown in Figure 2b. During the flocculation process, *C. vulgaris* cells came in contact with the flocculant and settled, and the flocculation efficiencies increased with increasing flocculation time. After 60 min of flocculation, the maximum flocculation efficiencies reached 98.67 ± 0.25%, 98.77 ± 0.31%, 97.37 ± 0.75%, and 93.53 ± 0.67% for $Al_2(SO_4)_3$, $AlCl_3$, $Fe_2(SO_4)_3$, and $FeCl_3$, respectively. This result demonstrated that a flocculation time of 60 min was adequate for harvesting *C. vulgaris*.

### 3.3. Stirring Speed

Stirring speed affects the binding degree of microalgae and flocculants. The flocculation efficiencies of *C. vulgaris* under various stirring speeds are shown in Figure 2c. When the stirring speed was low, the flocculation efficiency of *C. vulgaris* was low because a low stirring speed may lead to poor dispersion of flocculant in liquid and less contact between the positively charged flocculant and negatively charged *C. vulgaris* [29–31]. With the increase in stirring speed, the dispersion of flocculant, collision frequency, and contact intensity between *C. vulgaris* and flocculant increased, and the positively charged flocculant quickly neutralized the negatively charged *C. vulgaris* cells [31], resulting in a steady

increase in flocculation efficiency. At a stirring speed of 300 rpm, the optimal microalgae flocculation efficiencies for $Al_2(SO_4)_3$, $AlCl_3$, $Fe_2(SO_4)_3$, and $FeCl_3$ were $98.67 \pm 0.59\%$, $98.67 \pm 0.60\%$, $97.33 \pm 0.91\%$, and $93.47 \pm 0.71\%$, respectively. A further increase in stirring speed will lead to a reduction in flocculation efficiency because a high stirring speed will lead to excessive shearing, destroy the flocculant and microalgae flocs, and disperse the coagulated cells into the medium again [29,30].

### 3.4. Stirring Time

There was no significant difference in the flocculation efficiencies among the four tested flocculants (Figure 2d). With increasing stirring time, the flocculation efficiency first increases and then decreases (Figure 2d). A stirring time of 1 min resulted in the lowest flocculation efficiency, whereas 2 min of stirring time produced the maximum flocculation efficiency. Subsequently, with increasing stirring time, the flocculation efficiency decreased. A short stirring time was too brief for the flocculants to act, and coagulation was insufficient to form sufficient flocs, thus resulting in low flocculation efficiency [32]. With the prolongation of stirring time, coagulation was completed gradually, and the flocculation efficiency improved. In contrast, if the stirring time is too long, the flocs that are formed will be destroyed because of the shear force, the flocs will be dispersed into the suspension, and the flocculation efficiency will be reduced. Thus, 2 min was adopted as the optimal stirring time in the present study.

### 3.5. Flocculation pH

The pH can affect the surface charge of microalgal cells (zeta potential) and the degree of ionization of flocculants [23,33]. The results showed that acidic medium has the lowest flocculation efficiency for all the flocculants (Figure 2e). With increasing pH, the flocculation efficiencies increased gradually, and pH 8.0 was found to be the optimum pH to harvest *C. vulgaris*, with flocculation efficiencies reaching $98.77 \pm 0.31\%$, $93.50 \pm 0.87\%$, $98.73 \pm 0.85\%$, and $97.40 \pm 0.62\%$ for $AlCl_3$, $FeCl_3$, $Al_2(SO_4)_3$, and $Fe_2(SO_4)_3$, respectively. A further increase in pH value resulted in a decrease in flocculation efficiency.

Metal flocculants ($Al^{3+}$ and $Fe^{3+}$) can accept the negative charge on microalgal cells and destabilize the microalgae cell network by neutralizing the negative charge, leading to cell agglomeration and flocculation [34]. At different pH values, the surface charge instability of *C. vulgaris* cells affects the interaction of flocculant particles and flocculation efficiency [32]. Under acidic conditions, metal flocculants exist in the form of positively charged hydroxides, such as $Fe(H_2O)_6^{3+}$ and $Al(H_2O)_6^{3+}$ [35]. With the increase in the pH value, $Fe(H_2O)_6^{3+}$ and $Al(H_2O)_6^{3+}$ hydrolyses to form various polynuclear hydroxyl complexes with more positive charges, which can neutralize the more negatively charged *C. vulgaris* cells. Accordingly, the maximum flocculation efficiency was obtained at pH 8.0 in the present study. Under alkaline conditions (>8.0), $Fe(OH)_3$ and $Al(OH)_3$ precipitated, resulting in a decrease in flocculation efficiency [28].

Flocculation induced by pH variation has been studied in various microalgal strains [6,7,9,11,13,16]. Under alkaline conditions, *C. vulgaris*, *C. fusca*, *I. galbana*, *Chlorococcum* sp., *Scenedesmus* sp., *Scenedesmus spinosus*, *S. subspicatus*, *P. tricornutum*, and *Muriellopsis* sp. exhibited more than 87% flocculation efficiency [6,7,9,11,13,16,36,37]. In contrast, certain microalgae, such as *Chlorococcum nivale*, *Scenedesmus* sp., and *Chlorococcum ellipsoideum,* have been harvested (flocculation efficiencies of >90%) in acidic pH (1.5–5.0) suspensions [34]. In addition, *Chlorella* sp. [38] and *Scenedesmus* sp. [35] also exhibited the highest flocculation efficiencies under acidic pH conditions. The flocculation ability of different microalgae depends on culture age, physiological conditions, cell wall composition, and extracellular excretion compositions [23]. Therefore, different microalgal cultures require different pH conditions. It is important to note that the initial acid culture is expensive and requires an excessive amount of chemicals to neutralize the medium to achieve optimal harvesting efficiency. In the present study, pH 8.0 was found to be the optimum pH for harvest-

ing *C. vulgaris*, which was within the operating pH range for microalgal growth, thus significantly decreasing the chemical requirements.

### 3.6. Flocs Morphology

Figure 3 presents micrographs of raw and flocculated microalgae, demonstrating the effect of flocculants on the morphology of *C. vulgaris*. The raw cells without flocculants were dispersed fine and floc-free (Figure 3a), whereas after the addition of $Al_2(SO_4)_3$, $AlCl_3$, $Fe_2(SO_4)_3$, and $FeCl_3$, the surface of the microalgae was covered with the hydrolysates of $Fe^{3+}$ and $Al^{3+}$, which resulted in the formation of larger clumps or aggregations of the microalgae cells [39] (Figure 3b–e).

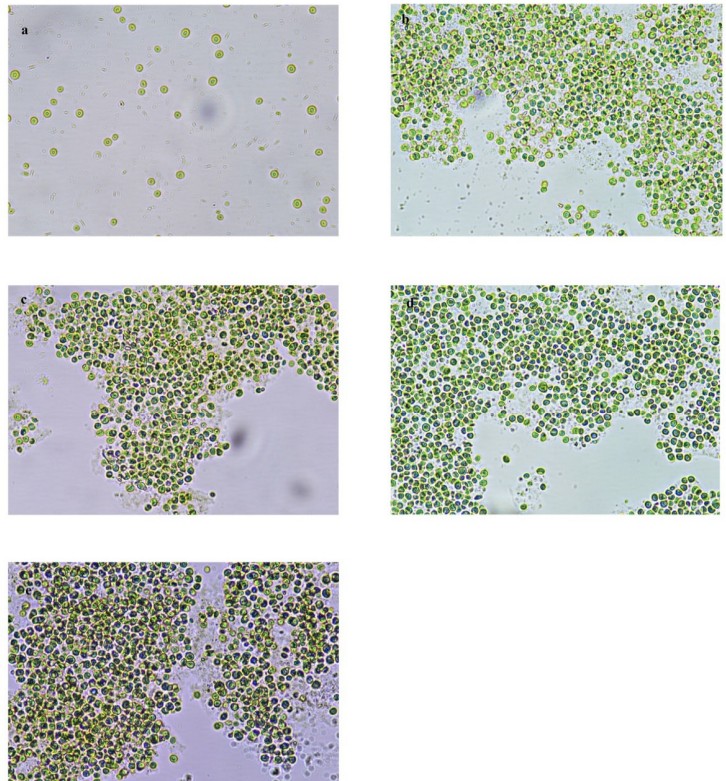

**Figure 3.** Microscopic images of *C. vulgaris* (**a**) raw (**b**) $Al_2(SO_4)_3$, (**c**) $AlCl_3$, (**d**) $Fe_2(SO_4)_3$, and (**e**) $FeCl_3$ flocculated at a magnification of 40×.

The microalgal cell SEM images (Figure 4) did not show obvious surface deformation or cell shrinkage, indicating that the cells almost did not decompose after neutralizing the charge with sulfate or chloride. Micrographs showed that the positive $Al^{3+}$ and $Fe^{3+}$ coagulants were attracted to the negative *Chlorella* cells to form flocs, consistent with the charge-neutralized primary coagulation mechanism [40]. A previous study reported that the cells of *Scenedesmus* sp. and *S. obliquus* were not damaged even by 17.1 g/L $Al_2(SO_4)_3$ during the flocculation process [8], while another study revealed that 150 mg/L $Al_2(SO_4)_3$ did not affect the cell morphology of *Picochlorum maculatum* MACC3 [41], which is in agreement with the results of the present study. In contrast, harvesting *Euglena gracilis* using NaOH at pH higher than 10 has been noted to cause complete rupture of cells [42]. These different findings can be explained by the different forms of microalgae (some microalgal strains have thick cell walls resistant to shock, including *P. maculatum*, *C. vulgaris* and *Scenedesmus* sp. However, others have no rigid cell wall, including *P. purpureum*) or the flocculant dose used.

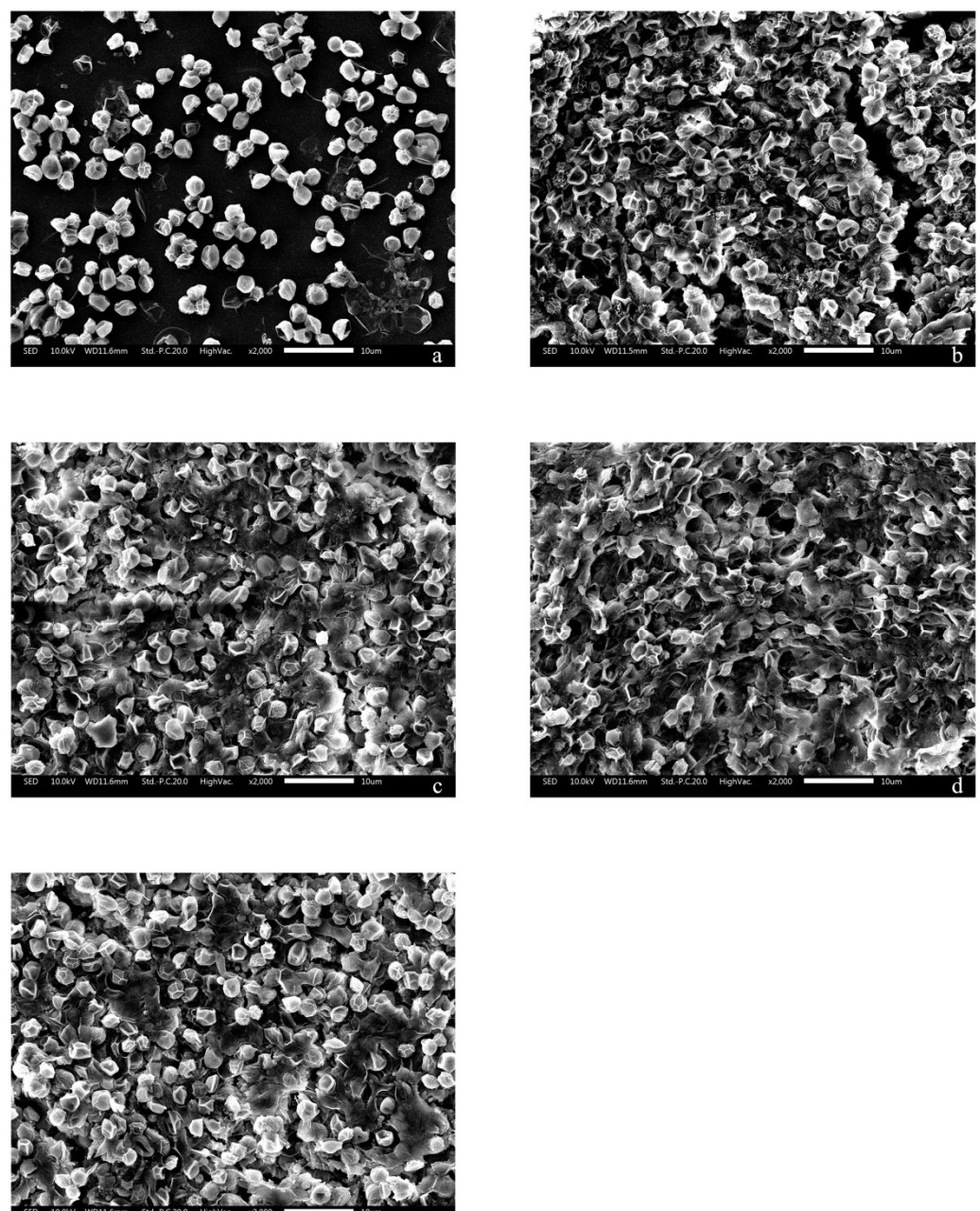

**Figure 4.** Scanning electron microscopy images of *C. vulgaris* (**a**) raw, (**b**) $Al_2(SO_4)_3$, (**c**) $AlCl_3$, (**d**) $Fe_2(SO_4)_3$, and (**e**) $FeCl_3$.

### 3.7. Biomass Composition

The ideal harvesting technique should not affect the quality of microalgae biomass and downstream processes [43]. Accordingly, the extraction of flocculation microalgae cells and proto-microalgae cells was compared in the present study (Table 2). The ratio of total proteins and carotenoids was not significantly affected by flocculants ($p > 0.05$), and the flocculating cells did not release the cell content into the medium during the collection process. Similar invariance in biomass composition after flocculation has also been reported in previous studies [17,44]. For carbohydrates, only $Fe_2(SO_4)_3$ decreased the total carbonhydrate contents. Aluminum salt ($Al_2(SO_4)_3$ and $AlCl_3$) decreased lipids contents. However, they only decreased slightly. Some reports indicate that the quality of biomass decreases after flocculation [13,45]. These results show that the flocculant has good industrial application prospects.

**Table 2.** Influence of different flocculants on biomass compositions. Data are reported with $+/-$ SD, and n = 3.

| | Total Carbohydrate | Total Protein | Total Lipid | Carotenoid |
|---|---|---|---|---|
| Control | $21.60 \pm 0.36$ [a] | $40.83 \pm 0.70$ [a] | $15.50 \pm 0.30$ [a] | $4.13 \pm 0.15$ [a] |
| $Al_2(SO_4)_3$ | $20.70 \pm 0.40$ [ab] | $40.37 \pm 0.21$ [a] | $14.93 \pm 0.25$ [b] | $3.93 \pm 0.06$ [a] |
| $AlCl_3$ | $20.77 \pm 0.57$ [ab] | $40.23 \pm 0.60$ [a] | $14.87 \pm 0.21$ [b] | $3.97 \pm 0.21$ [a] |
| $Fe_2(SO_4)_3$ | $20.37 \pm 0.31$ [b] | $40.63 \pm 0.42$ [a] | $15.00 \pm 0.36$ [ab] | $3.87 \pm 0.21$ [a] |
| $FeCl_3$ | $21.03 \pm 0.90$ [ab] | $40.37 \pm 0.67$ [a] | $15.10 \pm 0.20$ [ab] | $4.00 \pm 0.10$ [a] |

[a, b] Duncan's test, alpha = 0.05.

*3.8. Regrowth of Settled Flocs*

The flocculated cells immediately resumed growth after transferred to fresh culture medium (Figure 5a). The growth curves of raw and flocculated cells were basically the same, which showed that the cells did not decompose and that cell function was not affected. Both chitosan and polyacrylamide have been reported to have no effect on the regeneration of flocculated *Scenedesmus* cells, and the reused medium has no negative effect on microalgal growth [8]. Kim et al. [46] found that chemical stress of flocculants resulted in the slow *Scenedesmus* sp. cell growth and a decrease in biomass yield, while another study found that ferric salt treatment could not only maintain the growth of *Arthrospira platensis* but also improve biomass quality in the treated medium [44].

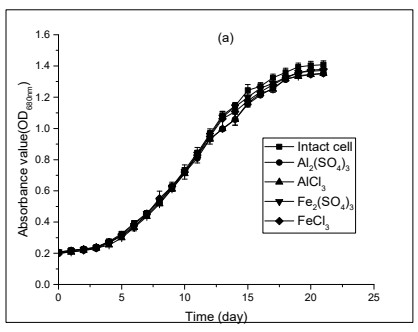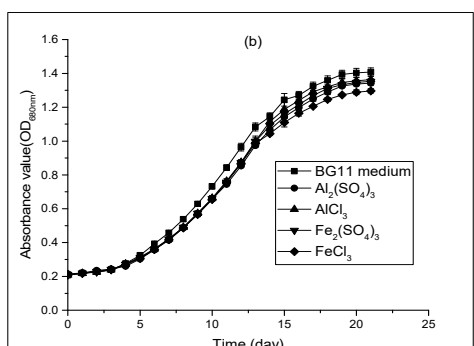

**Figure 5.** Cell growth curves of (**a**) algal cells in flocs and (**b**) intact cells in spent medium. Data are reported with $+/-$ SD, and n = 3.

*3.9. Reuse of Spent Medium*

The reuse of spent medium for microalgal cultivation can reduce the nutrient and water requirements, ultimately decreasing the microalgal production cost. The growth curves of *C. vulgaris* in the four flocculated media were similar to the growth curves in fresh BG11 medium, indicating that the four flocculated media could be reused (Figure 5b). Many studies have shown that reused water had no substantial effect on algal growth [8,44], whereas other reports have indicated improved growth profiles in recycled media with higher biomass (g/L) and lipid yield (%) [47,48]. However, in many cases, the residual flocculants in the aqueous medium often change the medium composition and adversely affect microalgae cells [9,18,49,50]. These varied effects (stimulatory, inhibitory, or neutral) of reused water on algae growth may be the result of many factors, including culture conditions, microalgal strain, harvesting method, reuse water scheme, etc.

**4. Practical Applications and Prospects**

*C. vulgaris* can be widely applied in many fields. Derived biofuels from microalgae lipids are considered to be one of the most efficient ways to harvest biofuels. Due to the renewable and clean properties, biofuel production has prospects for development [51]. In addition, *C. vulgaris* can be processed into food and feed and applied in sewage treatment [1].

With the development of flocculation technology, more factories will adopt inorganic salt flocculation to harvest *C. vulgaris*, which will greatly reduce the production cost of *C. vulgaris.*

## 5. Conclusions

This study compared the effective harvesting of *C. vulgaris* through flocculation using $Al_2(SO_4)_3$, $AlCl_3$, $Fe_2(SO_4)_3$, and $FeCl_3$ and investigated the flocculation performance. Low-dose sulfate salts (60 mg/L) caused efficient flocculation similar to high-dose chloride salts (100 mg/L). The maximum flocculation rate of 97.40–98.77% were achieved at optimum conditions (stirring 300 r/min for 2 min, pH 8.0, settled for 60 min) with 60 mg/L $Fe_2(SO_4)_3$, 100 mg/L $Fe_2(SO_4)_3$ or 100 mg/L $AlCl_3$. None of the four flocculants caused damage to the cells of *C. vulgaris*, and there were no significant differences in their biomass. In addition, the cells in flocs could effectively regenerate, and the four flocculated media could be recycled, thus reducing the production cost of microalgae.

**Author Contributions:** W.M.: Figure and Table, Writing—review and editing, Revise, Formal analysis. C.F. and F.G.: Raw data, Methodology. J.C.: Conceptualization, Data curation, Methodology, Writing—original draft. D.M.: Formal analysis, Writing—review and editing. All authors have read and agreed to the published version of the manuscript.

**Funding:** This work was funded by Jilin Province Agricultural Science and technology innovation project (KYJF2021JQ003), the Open Project Program of Key Laboratory of Marine Resource Chemistry and Food Technology (TUST), Ministry of Education (EMTUST-21-07), Tianjin Key Laboratory of Brine Chemical Engineering and Resource Eco-utilization (Tianjin University of Science & Technology) (BCERE202103), and Jilin Province Science and technology development plan project (20200403014SF, 20200602054ZP).

**Institutional Review Board Statement:** Not applicable.

**Informed Consent Statement:** Not applicable.

**Data Availability Statement:** The data and materials in this article are available from the corresponding authors on reasonable request.

**Conflicts of Interest:** No conflict, informed consent, or human or animal rights are applicable to this study.

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
