# Peer review of "Effective Chlorella vulgaris Biomass Harvesting through Sulfate and Chloride Flocculants"

_jmse, doi:10.3390/jmse11010047_

Round 1

Reviewer 1 Report

Line #16: ‘Chlorella vulgaris’ needs to be italicized.

Line #28: ‘Microalgae’ should be changed to ‘microalgae’ (capitalized form)

The flow of the first paragraph is not sound and convincing. The authors addressed explanation on Chlorella and then next suddenly changed the topic to harvesting difficulties. And the importance of research on microalgae harvesting is not fully highlighted so that it is not compelling.

Line #37: Harvesting cost à Instrumental cost for harvesting?

Line #47-49: The authors stated that flocculation is a superior method due to its efficiency and low-cost. However, why the industry has been widely using centrifugation method?

The authors stated that there is a concern of contamination by pollutants (Line #72-73) and water reuse is a method to reduce the cost (Line #75-76). However, can the water reuse avoid of the risk of contamination? How can the cost be reduced even if additional costs are required for purifying the water prior to reuse?

Line #139: ‘C. vulgaris’ needs to be italicized.

Representation of Figure 2 is confusing. Y-axis on flocculation efficiency should be identical among the figures.

As written in Line #203, measurement data on residual ion concentration will be helpful to bolster the author’s statement.

What would be the reason of the optimal flocculation pH 8 for Chlorella vulgaris in the present study although there is a report that Chlorella sp. showed the highest flocculation efficiency under acidic pH conditions (Line #270; reference 38)?

In microscopic (Figure 3) or SEM (Figure 4) observations, was there any damage of the cells probably caused by the conditions (dose, exposure time, pH, etc.) of flocculants?

Line #348: What were the results on the last column in Table 2?

Author Response

Dear Reviewer 1,

Thank you very much for your nice suggestion, which undoubtedly have largely improved our manuscript, even for our future research. According to your fine comments and suggestions, we have made carefully revisions to our original manuscript and give detailed responses to your questions. Your comments and our responses are included below. Our responses were marked in red color.

Question:

Line #16: ‘Chlorella vulgaris’ needs to be italicized.

Respond: Thank you for your fine suggestion. Chlorella vulgaris is italicized in the revised manuscript.

Line14,

Citation: Ma, W.; Feng, C.; Guan, F.; Ma, D.; Cai, J. Effective Chlorella vulgaris biomass harvesting through sulfate and chloride flocculants. J. Mar. Sci. Eng. 2022, 11, x.

Line 15-16,

In this study, sulfate (Al2(SO4)3 and Fe2(SO4)3) and chloride flocculants (AlCl3 and FeCl3) were used to harvest Chlorella vulgaris.

Line 367-368,

  1. Safi, C.; Zebib, B.; Merah, O.; Pontalier, P.Y.; Vaca-Garcia, C., Morphology, composition, production, processing and applications of Chlorella vulgaris: A review. Renew Sust Energ Rev 2014, 35, 265-278.

Question:

Line #28: ‘Microalgae’ should be changed to ‘microalgae’ (capitalized form)

Respond: Thank you for your fine suggestion. The “microalgae” have been changed to microalgae (capitalized form) in the revised manuscript.

Line 28,

Reactive microalgae which are regarded as promising and sustainable cell factories, are rich in high-value biocompounds, including zeaxanthin, β-carotene, lutein, agar, enzymes, polyunsaturated fatty acids, vitamins, amino acids, and various nutraceutical and pharmaceutical substances.

Question:

The flow of the first paragraph is not sound and convincing. The authors addressed explanation on Chlorella and then next suddenly changed the topic to harvesting difficulties. And the importance of research on microalgae harvesting is not fully highlighted so that it is not compelling.

Respond: Thank you very much. The first paragraph is revised according your fine suggestion.

Line 31-32,

Before these high-value biocompounds extracted from microalgae, an appropriate harvesting method is needed to concentrate the biomass.

Question:

Line #37: Harvesting cost à Instrumental cost for harvesting?

Respond: Thank you for your fine suggestion. Instrumental cost is important part of microalgal harvesting. However, flocculation did not need high-cost instrument, which could greatly decrease instrumental investing.

Question:

Line #47-49: The authors stated that flocculation is a superior method due to its efficiency and low-cost. However, why the industry has been widely using centrifugation method?

Respond: Due to our careless, the reasons for widely using of centrifugation method was insufficiency. Some reasons were added in the revised manuscript.

Line 41-44,

Among these methods, centrifugation is currently used in industry because of its high efficiency close to 100%. However, centrifugation has the limitations of high energy consumption as well as operating cost 5 and mainly used in high-priced goods.

Question:

The authors stated that there is a concern of contamination by pollutants (Line #72-73) and water reuse is a method to reduce the cost (Line #75-76). However, can the water reuse avoid of the risk of contamination? How can the cost be reduced even if additional costs are required for purifying the water prior to reuse?

Respond: Thank you for your fine suggestion. Insufficient discussion about the contamination is in the original manuscript. Related discussion was added in the revised manuscript. Iron was added at the first batch, however, as there was residual iron in the recycled medium, iron was not required in the followed culture. The growth of C. vulgaris could utilize some of the residual iron in the recycled culture medium, which could reduce some contamination and cost.

Line 130-132,

The spent medium pH and nutrient concentration were adjusted to standard BG11 medium, and fresh algal cells were inoculated into the spent medium.

Question:

Line #139: ‘C. vulgaris’ needs to be italicized.

Respond: Thank you for your fine suggestion. “C. vulgaris” was italicized in the revised manuscript.

Line 105-107,

Before flocculation, the C. vulgaris suspension was diluted with BG11 medium to an OD680 (optical density measured at 680 nm) of 1.2 (approximately 708 mg dry weight/L) to obtain comparable initial conditions between different runs.

Line 137-138,

The C. vulgaris cells before and after harvesting were observed under a brightfield microscope at 400× magnification (H550S Nikon, Japan)

Question:

Representation of Figure 2 is confusing. Y-axis on flocculation efficiency should be identical among the figures.

Respond: Thank you for your fine suggestion. However, to increase the immediacy of contrast, the Y-axis on flocculation efficiency is varied in different figures.

Question:

As written in Line #203, measurement data on residual ion concentration will be helpful to bolster the author’s statement.

Respond: Thank you for your fine suggestion. Data about residual ion concentration will greatly helpful to bolster our statement. But the results of regrowth of settled flocs and reuse of spent medium could also support our statement. We will measure the residual ion concentration in future.

Question:

What would be the reason of the optimal flocculation pH 8 for Chlorella vulgaris in the present study although there is a report that Chlorella sp. showed the highest flocculation efficiency under acidic pH conditions (Line #270; reference 38)?

Respond: Thank you for your fine suggestion. The flocculation ability of different microalgae depends on the culture age, physiological conditions, cell wall composition, and extracellular excretion compositions. There are some studies found varies optimum pH for Chlorella flocculation. For example, about pH 10 have higher flocculation efficiency than pH 9.4 in Chlorella sp. 725, Chlorella sp. 615 and Chlorella sp. 442 [1]. pH 10.5-pH 12 have higher flocculation efficiency than pH 10 for Chlorella vulgaris [2]. pH 7.0 showed the maximum harvesting efficiency for Chlorella vulgaris [3] and Chlorella pyrenoidosa [4].

[1] Yang, F., et al. (2016). "High pH-induced flocculation of marine Chlorella sp for biofuel production." Journal of applied phycology 28(2): 747-756.

[2] Saul Garcia-Perez, J., et al. (2014). "Influence of magnesium concentration, biomass concentration and pH on flocculation of Chlorella vulgaris." Algal Research-Biomass Biofuels and Bioproducts 3: 24-29.

[3] Salama, E.-S., et al. (2015). "Application of acid mine drainage for coagulation/flocculation of microalgal biomass." Bioresource Technology 186: 232-237.

[4] Jiang, J., et al. (2021). "Harvesting of Microalgae Chlorella pyrenoidosa by Bio-flocculation with Bacteria and Filamentous Fungi." Waste and Biomass Valorization 12(1): 145-154.

Line 269-271,

The flocculation ability of different microalgae depends on culture age, physiological conditions, cell wall composition, and extracellular excretion compositions 23.

Question:

In microscopic (Figure 3) or SEM (Figure 4) observations, was there any damage of the cells probably caused by the conditions (dose, exposure time, pH, etc.) of flocculants?

Respond: Thank you for your fine suggestion. It is quite important to preventing cell membrane damage during harvesting avoids the leakage of compounds of interest into the growth medium, assuring overall yield and quality of the biomass harvested. There are many factors could result cell damage. For example, shear stress from stirring could cause cell damage of microalgae. Increasing the flow rate and time of stirring could not only cause cell damage of microalgae, but also could waste enormous energy. pH can affect the surface charge of microalgal cells (zeta potential) and the degree of ionization of flocculants. Extreme pH can lead to microalgal cell damage. Osmotic pressure results from flocculants will also damage microalgal cells. In this study, there is no obvious surface deformation or cell shrinkage.

Alternatively, microalgal cells could sense the remaining flocculants and adapt their metabolism to limit damage to their cell membranes.

Line 283-286,

after the addition of Al2(SO4)3, AlCl3, Fe2(SO4)3, and FeCl3, the surface of the microalgae was covered with the hydrolysates of Fe3+ and Al3+, which resulted in the formation of larger clumps or aggregations of the microalgae cells 39 (Fig. 3(b)–(e)).

Line 289-291,

The microalgal cell SEM images (Fig. 4) did not show obvious surface deformation or cell shrinkage, indicating that the cells almost did not decompose after neutralizing the charge with sulfate or chloride.

Question:

Line #348: What were the results on the last column in Table 2?

Respond: Due to our careless, the last column in Table 2 is missed in our original manuscript. It is Carotenoid.

Line 345,

Total Carbohydrate

Total Protein

Total Lipid

Carotenoid

Reviewer 2 Report

Reduce the plagrism

Overall paper is alright so requesting you to kindly go through the language of the paper which requires improvement in overall paper after making changes paper is ready for submission

Author Response

Dear Reviewer 2,

Thank you very much for your nice suggestion, which undoubtedly have largely improved our manuscript, even for our future research. According to your fine comments and suggestions, we have made carefully revisions to our original manuscript and give detailed responses to your questions. Your comments and our responses are included below. Our responses were marked in blue color.

Question:

Reduce the plagrism

Respond: Thank you for your suggestion. During the manuscript writing process, we have referred to many reported papers. we have carefully decreased the

Line 13-16,

Efficient microalgae harvesting is a great challenge hindering diverse industrial applications of microalgae. Flocculation is regarded as an effective and promising technology for microalgae harvesting. In this study, sulfate (Al2(SO4)3 and Fe2(SO4)3) and chloride flocculants (AlCl3 and FeCl3) were used to harvest Chlorella vulgaris.

Line 37,

harvesting cost is approximately 30% of the total microalgal production cost 4.

Line 58-59,

Different flocculants showed different flocculation efficiency on freshwater and marine microalgal strains, which is listed in Table 1.

Line 60,

If there is no strict demand for the supernatant, inorganic flocculants is a good choice

Line 77-80,

A few studies have been conducted on the regeneration of microalgae from flocculated and sterilized or filtered waste media such as Scenedesmus sp., C. vulgaris, C. fusca, Muriellopsis sp., Scenedesmus subspicatus, and Nannochlropsis oculate.

Line 113-114,

  1. vulgaris cells were mixed using a magnetic stirrer to ensure complete dispersal of the flocculants after adding the flocculant.

Line 188-189,

When flocculant concentration reached certain value, the metal cations completely neutralized the surface negative charge on the microalgal cells.

Line 256-259,

With the increase in the pH value, Fe(H2O)63+ and Al(H2O)63+ hydrolyses to form various polynuclear hydroxyl complexes with more positive charges, which can neutralize the more negatively charged C. vulgaris cells.

Line 259-261,

Accordingly, the maximum flocculation efficiency was obtained at pH 8.0 in the present study. Under alkaline conditions (> 8.0), Fe(OH)3 and Al(OH)3 precipitated, resulting in a decrease in flocculation efficiency.

Line 314-315,

The flocculated cells immediately resumed growth after transferred to fresh culture medium

Line 334-336,

These varied effects (stimulatory, inhibitory, or neutral) of reused water on algae growth may be the result of many factors, including culture conditions, microalgal strain, harvesting method, reuse water scheme, etc.

Line 339-340,

Compared with traditional fossil fuels, biofuel production has prospects for development 51.

Question:

Overall paper is alright so requesting you to kindly go through the language of the paper which requires improvement in overall paper after making changes paper is ready for submission

Respond: Thank you for your fine suggestion. This manuscript has been language revised by English expert.

Round 2

Reviewer 1 Report

The manuscript seems well-revised according to the comment in the first round. Apparently, the fact that the flocculation characteristics vary from strain makes the real application difficult. The author should emphasize the strain used in the study with the fact above in the conclusion part. 

Line #338: Vulgaris -> vulgaris

Line #339-340: The sentence "Compared with~" is not clear enough to deliver what the authors want to tell.

Line #345 (Table 2): Statistical analysis results should be represented next to the data in Table 2 to determine if there is any significant difference between the flocculant types. Otherwise, it needs to be stated in the footnote space. 

It seems necessary to address in the Conclusion part what the best flocculation condition was. And it would be helpful for the readers to find what would be the limitations to overcome in future studies.

Author Response

Dear Reviewer 1,

Thank you very much for your nice suggestion, which undoubtedly have largely improved our manuscript, even for our future research. According to your fine comments and suggestions, we have made carefully revisions to our original manuscript and give detailed responses to your questions. Your comments and our responses are included below. Our responses were marked in red color.

Question:

The manuscript seems well-revised according to the comment in the first round. Apparently, the fact that the flocculation characteristics vary from strain makes the real application difficult. The author should emphasize the strain used in the study with the fact above in the conclusion part. 

Respond:Thank you for your fine suggestion. The special microalgal strain (C. vulgaris) was added to the revised manuscript.

Line 350-351,

This study compared the effective harvesting of C. vulgaris through flocculation using Al2(SO4)3, AlCl3, Fe2(SO4)3, and FeCl3 and investigated the flocculation performance.

Question:

Line #338: Vulgaris -> vulgaris

Respond:Thank you for your fine suggestion. Due to our careless, the word is misspelled in the original manuscript. We have corrected it.

Line 338,

  1. vulgaris can be widely applied in many fields.

Question:

Line #339-340: The sentence "Compared with~" is not clear enough to deliver what the authors want to tell.

Respond:Sorry for our confusion description, the sentence is revised in the new manuscript.

Line 343-344,

Due to the renewable and clean properties, biofuel production has prospects for development

Question:

Line #345 (Table 2): Statistical analysis results should be represented next to the data in Table 2 to determine if there is any significant difference between the flocculant types. Otherwise, it needs to be stated in the footnote space.

Respond:Thank you for your fine suggestion. Statistical analysis was added in the revised manuscript.

Line 310-312,

For carbohydrates, only Fe2(SO4)3 decreased the total carbonhydrate contents. Aluminum salt (Al2(SO4)3 and AlCl3) decreased lipids contents. However, they only decreased slightly.

Line 315,

Table 2. Influence of different flocculants on biomass compositions. Data are reported with +/− SD, and n = 3.

Total Carbohydrate

Total Protein

Total Lipid

Carotenoid

Control

21.60±0.36 a

40.83±0.70 a

15.50±0.30 a

4.13±0.15 a

Al2(SO4)3

20.70±0.40 ab

40.37±0.21 a

14.93±0.25 b

3.93±0.06 a

AlCl3

20.77±0.57 ab

40.23±0.60 a

14.87±0.21 b

3.97±0.21 a

Fe2(SO4)3

20.37±0.31 b

40.63±0.42 a

15.00±0.36 ab

3.87±0.21 a

FeCl3

21.03±0.90 ab

40.37±0.67 a

15.10±0.20 ab

4.00±0.10 a

a, b Duncan’s test, alpha=0.05

Question:

It seems necessary to address in the Conclusion part what the best flocculation condition was. And it would be helpful for the readers to find what would be the limitations to overcome in future studies.

Respond:According to your fine suggestion, we have revised the manuscript. Which will greatly be increased our manuscript.

Line 351-353,

The maximum flocculation rate of 97.40-98.77 % were achieved at optimum conditions (stirring 300 r/min for 2 min, pH 8.0, settled for 60 min) with 60 mg/L Fe2(SO4)3, 100 mg/L Fe2(SO4)3 or 100 mg/L AlCl3

Reviewer 2 Report

Manuscript has been well-revised according to the previous comments

I recommend to accept in present form.

Round 3

Reviewer 1 Report

The manuscript seems well-revised according to the comment in the previous rounds. The Abstract section has been changed more concise. 

Just one minor comment: According to the caption, the author should mark (a), (b), (c), (d), and (e) outside of each figure in Figure 3. The marks are not clearly seen.